# Enhancing Building Point Cloud Reconstruction from RGB UAV Data with Machine-Learning-Based Image Translation

**DOI:** 10.3390/s24072358

**Published:** 2024-04-08

**Authors:** Elisabeth Johanna Dippold, Fuan Tsai

**Affiliations:** 1Department of Civil Engineering, National Central University, 300, Zhongda Rd., Zhongli, Taoyuan 32001, Taiwan; elisabeth.dippold@gmail.com; 2Center for Space and Remote Sensing Research, National Central University, 300, Zhongda Rd., Zhongli, Taoyuan 32001, Taiwan

**Keywords:** structure-from-motion, cross-sensor, image-to-image translation, conditional GAN, vegetation segmentation

## Abstract

The performance of three-dimensional (3D) point cloud reconstruction is affected by dynamic features such as vegetation. Vegetation can be detected by near-infrared (NIR)-based indices; however, the sensors providing multispectral data are resource intensive. To address this issue, this study proposes a two-stage framework to firstly improve the performance of the 3D point cloud generation of buildings with a two-view SfM algorithm, and secondly, reduce noise caused by vegetation. The proposed framework can also overcome the lack of near-infrared data when identifying vegetation areas for reducing interferences in the SfM process. The first stage includes cross-sensor training, model selection and the evaluation of image-to-image RGB to color infrared (CIR) translation with Generative Adversarial Networks (GANs). The second stage includes feature detection with multiple feature detector operators, feature removal with respect to the NDVI-based vegetation classification, masking, matching, pose estimation and triangulation to generate sparse 3D point clouds. The materials utilized in both stages are a publicly available RGB-NIR dataset, and satellite and UAV imagery. The experimental results indicate that the cross-sensor and category-wise validation achieves an accuracy of 0.9466 and 0.9024, with a kappa coefficient of 0.8932 and 0.9110, respectively. The histogram-based evaluation demonstrates that the predicted NIR band is consistent with the original NIR data of the satellite test dataset. Finally, the test on the UAV RGB and artificially generated NIR with a segmentation-driven two-view SfM proves that the proposed framework can effectively translate RGB to CIR for NDVI calculation. Further, the artificially generated NDVI is able to segment and classify vegetation. As a result, the generated point cloud is less noisy, and the 3D model is enhanced.

## 1. Introduction

Image-to-image translation techniques based on Generative Adversarial Networks (GANs) are a solution to filling data gaps [1]. Among the available GAN algorithms, the conditional GAN called Pix2Pix [2] can generate a corresponding image output based on input image information. In addition, the modified two-view Structure from Motion (SfM) algorithm can generate a 3D point cloud from two 2D images [3,4]. The process of reconstructing a 3D model with SfM involves taking images from different viewpoints (camera motion). These viewpoints can be related to each other by feature detection and matching [5]. The process of feature detection [6] and matching is crucial [7] to successfully estimating the pose and triangulating the features [8]. However, the performance of three-dimensional (3D) reconstruction from images is affected by many factors. Therefore, integrating additional data is often a practical necessity for better SfM-based 3D reconstruction.

In general, the need for more data can be tackled by the integration of different sensor data, by additional measurements, by data fusion or by new sensors. Similar machine learning approaches have also been successfully used in construction and civil engineering applications, such as using ultra-wideband sensors for structural health monitoring [9], and further enhanced by extreme gradient boosting decision trees [10]. Likewise, the machine learning analysis of fiber-optic sensor data has also been used to monitor structural health by measuring strain and crack distributions [11], as well as perform the automatic and real-time monitoring of pipeline corrosion [12].

For airborne and close-range imagery, different approaches have been proposed and proven effective, including integrating Airborne Lidar (ALS) and Unmanned Aerial Vehicles (UAVs) within a SfM framework for plantation forest management [13]; the prediction of biomass with RGB and height measurements [14]; multiplatform SfM applications for single high-resolution topography (HRT) with RGB SfM aerial and terrestrial images; and the use of a Global Navigation Satellite System (GNSS) and Terrestrial Laser Scanner (TLS) [15]. When reconstructing 3D models of static targets (e.g., buildings) using SfM algorithms, vegetation is often considered as noise and may decrease the performance significantly. Therefore, identifying vegetation and adjusting the SfM operations accordingly should improve the 3D reconstruction result of static targets. In general, the remote sensing applications of multispectral images beyond visible RGB bands are used to identify vegetation and other ground covers [16,17,18,19], or to distinguish different kinds of vegetation like viticulture (vine) [20] or rice [21]. However, some sensors, especially UAV cameras, are popular data acquisition sources for SfM reconstruction that provide RGB bands only, meaning that the detection of vegetation or other natural properties is limited. Consequently, supplementing additional information with these types of images may help improve the performance of SfM-based reconstructions.

The objective of this study was to improve the performance of 3D point cloud generation using UAV or close-range RGB images. To achieve this, the integration of new techniques like image-to-image translation to overcome the lack of NIR data is necessary as it can help better detect and mask vegetation areas on the images, which, as discussed above, are considered noise and may decrease the effectiveness of point cloud generation. In addition to the development, training and validation of a machine learning model for image-to-image translation and a point cloud generation framework, this study also tests the developed approach with different datasets and evaluates the results qualitatively and quantitatively.

## 2. Related Work of GAN, Vegetation Segmentation and SfM

Artificial Intelligence (AI) in general and Machine Learning (ML) in particular have gained more and more momentum in recent years. This momentum can be quantified by the great number of different techniques addressing various problems. In general, supervised AI ML techniques feed a model by providing input and then prediction. The predicted output can then be compared to the expected output. Afterwards, users can update the model to generate a different predictive output [22,23,24]. In contrast, Generative Adversarial Networks (GANs) are a group of AI ML unsupervised techniques. In GANs, a generator produces fake samples and a discriminator is used to distinguish between real (input) and fake (generated) samples. This interaction between generator and discriminator can be described as adversarial, with two models competing against each other [1].

GANs are primarily used for image translation, including translation from 2D to 3D, image-to-image translation or are specialist for one topic. A variety of GAN modifications and applications have been proposed, including dealing with unmatched datasets (Distance GAN [25]) and samples of different domains (Δ-GAN [26]), as well as generating a range of possible outcomes like the range from day to night (BicycleGAN [27]). The translation from 2D to 3D can be accomplished by depth estimation based on a single image input (T2Net [28]), or directly to a 3D model (3D-RecGAN [29]) or 3D shape (PrGAN [30]).

Image-to-image translation can be applied to increase the resolution (AffGAN [31]), to generate thermal images (IR2VI [32]), to explore different color spaces (manifold-WGAN [33]), contrasts (ContrastGAN [34]), image conditions (DNA-GAN [35]), energies (EBGAN [36]) or cross domains (DTN [37]). Some studies have used GAN for image-to-image translation to generate visible-like images from synthetic aperture radar (SAR) (SAR-GAN [38]) or Himawari for green band generation [39]. For images without any NIR channel, image-to-image translation algorithms can be used to generate a pseudo-NIR band in order to collect information about vegetation. This is particularly useful for vegetation [40], crop [41], forest [42] or fruit [43]-related applications.

The feature detection step of the SfM algorithm can benefit from the adjustment of the initial key points and image alignment with the application of a neural network [44]. Alternatively, an adjustment to the camera setup, like Multi-Camera SfM, can increase the efficiency, robustness and accuracy of the SfM algorithm [45]. In contrast, the limitation of the data may reduce the manual labor and computational resources used while successfully extracting building information from satellite imagery resources [46].

In a different study, object-based segmentation was employed to identify palm trees in order to monitor the growth and development of every single tree on a plantation [47]. The segmentation of the generated or sensed point cloud can be achieved instantly and unsupervised, for instance, as part of the reconstruction analysis [48]. Extracting vegetation directly from RGB UAV images is possible, but the results are often limited and have certain conditions [49,50].

When performing dense matching for point cloud generation, a commonly adopted approach is eliminating occlusions and other unwanted objects (such as vegetation) to focus on the target (e.g., buildings) only. This can be achieved by semantic-guided reconstructions (SGRs) [51]. Another approach is separating extracted features into different groups to reduce the interference of vegetation change in target detection [52]. In this regard, the application of Semantic Flow Field-guided DSM Estimation (SFFDE) can generate an elevation map (with vegetation) on the one hand, and on the other hand, filter vegetation out to generate a building mask [53].

In addition, the correct segmentation of vegetation (and removing them) is an important issue. However, most close-range and UAV images usually consist of no NIR channels, which makes vegetation segmentation a challenging task. Therefore, this study developed a two-stage framework for vegetation-segmented two-view SfM. The proposed two-stage framework can simulate color infrared images from original RGB bands to be used for vegetation segmentation with a GAN-based image translation algorithm. As a result, the vegetation segmentation will improve the feature detection and image matching in the two-view SfM operation and produce better 3D point cloud data of buildings or other static targets.

## 3. Methods, Materials and Experiment

### 3.1. Methods and Strategy

The two-stage workflow is roughly illustrated in Figure 1 and can be separated into four key ideas or into two stages. Firstly, color infrared (CIR) is generated from RGB data by image-to-image translation. The multispectral dataset is then used to calculate the NDVI to segment vegetation. This is performed so that vegetation occludes the target and so that the vegetation is highly dynamic. In addition, vegetation is a weak feature (in the image-matching phase of 3D reconstruction) because of its homogenous texture and lack of individuality. As a result, feature detection and matching are performed with respect to the NDVI classification. After that, the point cloud generation with pose estimation and triangulation is based on the remaining features.

The first stage covers the first two key ideas introduced in Figure 1. This includes the application of image-to-image translation to generate CIR from RGB. The first step consists of the pre-processing of the training data, combining very high-resolution satellite images with camera images. Next, color infrared (CIR) is created for training. Then, testing and validation is performed to choose the best model and predict the CIR from the remaining RGB data (Figure 2). The second stage covers the last two key ideas introduced in Figure 1, which involves the calculation of the NDVI and feature detection and matching with the NDVI, as illustrated in Figure 3. This includes the removal of features classified as vegetation based on the NDVI. Finally, a sparse point cloud is generated with respect to the available sensor data.

#### 3.1.1. The first Stage

The conditional machine learning image-to-image translation technique Pix2Pix (Figure 2b) takes pair-wise input in order to train the translation from one image to another image. Therefore, the input needs to be preprocessed first (Figure 2a). The input consists of an RGB and NIR dataset from cameras and a satellite. The cameras used in the EPFL dataset are a Nikon D90 (Nikon. Tokyo, Japan)for the RGB and Canon T1i (Canon, Tokyo, Japan), with a filter used for both cameras. The satellite imagery applied in this study is Pleiades 1B tri-stereo with a multispectral as well as panchromatic band. Firstly, the datasets are divided into training, testing and validation, and then processed. The training input size of the Generator (U-Net) applied in this study (Figure 2b) is 265 × 265 × 3. The generator generates CIR from RGB. Then, the discriminator decides if the fake is better than the ground truth by applying a PatchGAN and Adam optimizer. The desired band (Figure 2a), NIR, is generated from CIR, which is composed of NIR, red and green. After training, model selection is based on the FID and confusion matrix, as well as the overall robustness of the FID graph. Then, the testing and validation of the selected model with CIR, NIR and the NDVI, cross-sensor-wise, are performed. The output of the first stage is a CIR composite image predicted by the selected model.

#### 3.1.2. The second Stage

The Structure from Motion (SfM) with Normalized Differential Vegetation Index (NDVI) segmentation operation is summarized in Figure 3. The first step consists of segmentation and classification with the NDVI based on the Pix2Pix-generated CIR image (Figure 3, green). The next step is the feature detection with multiple feature detector operators applied (Figure 3, orange). Afterwards, features classified as vegetation are removed and masked out, then feature matching is performed. The final step is pose estimation with the remaining features and triangulation to generate the point cloud.

### 3.2. Materials

#### 3.2.1. RGB-NIR Training Dataset

The RGB-NIR scene dataset from EPFL [54] consists of 9 categories with approximately 50 scenes as RGB and NIR per category (Table 1). The versatility of the close range and large scale as well as the various scenes make the EPFL dataset appropriate for many different applications and training scenarios.

#### 3.2.2. UAV Dataset for Testing

The developed vegetation segmentation two-view SfM reconstruction framework was tested on a UAV image dataset captured using a DJI P3Pro UAV equipped with a RGB camera. The camera senses a 5.1K video resolution at a rate of 50 frames per second. The scene shows buildings (the target) among trees in front and a city skyline in the background. The view is mainly blocked by several tall trees in the foreground, which makes the reconstruction challenging. Firstly, the trees occlude part of the buildings and cast shadows. Secondly, the tree features detected in this area are highly similar, so that miss-matches are likely to occur. Finally, nature is in general more dynamic, so that slight changes caused by wind or sunshine, and weather in general, have a greater impact and may affect the feature matching significantly. However, the UAV is limited to RGB images, so an image translation operation to simulate CIR images would overcome this limitation. This would enable the developed algorithm to calculate the NDVI with the NIR band extracted from the simulated CIR data for vegetation segmentation, which would improve the feature matching and 3D point cloud generation performance.

#### 3.2.3. VHR Satellite Imagery for Training and Validation

The tri-stereo pair of Pleiades-1B very high-resolution satellite imagery shown in Figure 4 and the pairs detailed in Table 2 were used for the training and validation of the image-to-image translation. Pleaides captures images with a resolution of 0.5 m for the panchromatic (PAN) and 2.8 m for the multispectral (MS) bands. The scene exhibits a wide range of land cover types, including water streams, several highways, buildings, as well as a vegetation-covered mountain chain leaning into the city area. The original Pleiades images consist of the RGB and NIR band, which can be used to train and check (ground truth) the data.

#### 3.2.4. Study Target for Validation

The target building is shown in Figure 5a. Validation with respect to the target is conducted in three ways. Firstly, visual validation is performed with a strict focus on the target, the stadium (Figure 5b). The area consists of 9 tiles in three rows; each tile has 256 × 256 pixels. Secondly, the terrain section (c) consists of the lower portion of the target (a) with a size of 2320 × 463 pixels. Thirdly, a histogram-based case study including a visual validation is performed.

The prediction of this target within its environment is challenging for several reasons. The vegetation areas shown in red (b) or in brighter pixels in (c) and (d) consisting of single trees, trees in rows and randomly dense trees. In addition, the green court inside the stadium, green around the stadium and lawn between the streets are also vegetated areas. The non-vegetated areas are mainly streets and highways, the stadium with highly reflective solar panels and a variety of different commercial and residential buildings.

### 3.3. Experiments

#### 3.3.1. Preprocessing

The first stage of the framework aims to generate CIR from RGB with image-to-image translation. The preprocessing of the camera (EPFL dataset) differs slightly from that of the satellite imagery. The satellite imagery is normalized from 16-bit to 8-bit. Then, the panchromatic band is used to pansharp the multispectral imagery. The Coupled Nonnegative Matrix Factorization (CNMF) algorithm [55] is used to perform the pansharpening process.

Then, all images for training are processed by slicing and adding padding to generate 256 × 256 × 3 RGB and CIR tiles. Firstly, CIR is created out of the NIR, red and green bands and the size is adjusted so that the images are dividable by 256 with padding. The number of tiles for the PAN are sufficiently large enough, being 7221, 6888 and 7052, within the first run just by slicing (Table 3). However, the MS and EPFL need to increase the number of tiles for training, testing and validation (Table 3). Therefore, morphological operators (Figure 6) are applied to increase the number of tiles. Tiles 1 are the whole image morphed with flipping, mirroring and mirroring the flipped image (Figure 6a–d). The second row (Tiles 2) of morphological changes are rotations to left and right including the mirroring of each tile, respectively (Figure 6e–i).

#### 3.3.2. Training and Model Selection

Each training model is firstly tested with the Fréchet Inception Distance (FID), which focuses on similarity [56]. This study adopts the idea of [39] and extends the model selection process. Firstly, the training is run over 200 epochs and every 25th epoch’s model is saved (Table 4 Training 9). This is evaluated with the FID and Generator Loss (Figure 7) to detect the most suitable model range based on the FID and Generator Loss (Loss GEN). In the proposed framework, a random 10% or 26 × 26 of each NIR patch is used for FID calculations and the Loss GEN is captured using the loss function during training. The graph of Training 9 shown in Figure 7 suggests that besides the variation, there is a need to search for the best-performing model. In this case, the most appropriate model range is located around epoch 75 (red rectangle in Figure 7). As a result, further training is limited to 100 epochs. These training attempts limited to the EPFL dataset are summarized in Table 4. As listed in the table, the training time for 9 categories (Training 9) was long. However, reducing Training 1 to one category and Training 2 to two categories with respect to the target, which consists of vegetation and non-vegetated areas, reduced the training time significantly. A further reduction to 16 images per category or 3840 tiles reduced the time to approximately 6 h per category.

The training and testing for model selection are summarized in Table 5. The performance is evaluated for every epoch with FID and the graph is displayed in Figure 8 for each run, respectively. FID testing is conducted consistently with 6 images (EPFL) per category or 1440 tiles by randomly selecting 10% of each NIR patch. The results of the FID are plotted with different colors in Figure 8.

The training for the model selection is divided into four training runs, including using the entire EPFL dataset, using only the Streets category, using the Streets and Forests categories, and finally, combining the Streets and Forests of EPFL with the MS and PAN of the Pleiades dataset. The times needed for training with the different categories described are recorded and listed in Table 5. Furthermore, the numbers of categories applied and the resulting time, as well as the testing set, are listed. Likewise, the table also lists the number of categories with respect to the dataset, EPFLE RGB-NIR, Pleiades MS or Pleiades PAN. Pleiades MS and Pleiades PAN consist of three images in total, so have three categories each.

This study balanced out testing and validation over all categories, rather than following the idea of a percentage split. The first run used all 9 categories by splitting and completely utilizing the EPFL dataset in training, testing and validation. Training run 9 (Overall) processed 22,260 tiles, took more than 50 h and covered all 9 categories of the EFPL dataset. The FID evaluation of the overall EPFL is plotted in orange in Figure 8 and exhibits an unstable performance. The lowest FID of 1.889 is still much higher than all the others. As a result, the overall EPFL is excluded from further evaluation.

The other three training trials are much more similar and stable in terms of their FID performance, so further evaluation is conducted in order to determine the most appropriate model for our application. The Confusion Matrix (CM) evaluates the performance of the NDVI classification. This study uses four parameters of the CM to evaluate the remaining training trials (Table 6). The epochs with the best FID and CM are indicated by columns with respect to the training run color (Figure 8). Further, the evaluation with the confusion matrix is used to select the best model from the remaining three training trials (Table 6). The lowest FID values of each training are 1.889 (Overall), 1.132 (Street), 0.839 (Street and Forest) and 1.663 (Street, Forest, MS and PAN). The best model selection is marked for each training, except the overall for FID and CM evaluation. The training model selection with FID and CM indexes is summarized in Table 6. The parameters used to evaluate the performance of the NDVI with the confusion matrix are the accuracy, precision, F1-score and kappa coefficient. Further, the model with the lowest FID is compared with the model with the best classification results. Interestingly, epoch 89 of the Street training performance is the best, both for the FID and CM. In addition, Street is the best in terms of accuracy, precision and F1-score. However, the kappa coefficient shows that the result is less robust and more random. The repeatability of the prediction is of great importance in this study.

Street seems to perform the best in terms of accuracy and FID similarity. However, the training with Street, Forest, MS and PAN is much more stable and provides more robust results. Therefore, epoch 64 with a kappa coefficient of 0.7767 seems to be the most reliable model for the prediction of RGB2CIR in this study. As a result, epoch 64 of Street, Forest, MS and PAN training is considered to be the most appropriate in terms of robustness, accuracy and repeatability, and is therefore selected for prediction and further validation and application.

### 3.4. Vegetation Segmentation and Structure from Motion

The second stage, as introduced in Figure 3, focuses on the generation of a sparse point cloud with vegetation-segmentation-driven two-view Structure from Motion (SfM). Firstly, the Normalized Differential Vegetation Index (NDVI) is calculated for segmentation and classification. Then, feature detection is performed and points identified as vegetation are removed. Outliner removal and inlier matching with masking are further performed. Finally, pose estimation, triangulation and sparse point cloud generation are conducted.

This study uses, as mentioned, Pleiades satellite imagery as one of the datasets. On the one hand, it is used for training, and on the other hand, it is used for testing and validation. The color infrared (CIR) image was further pansharpened with the panchromatic (PAN) band, as shown in Figure 9, to produce NDVI images with higher spatial details. Pansharpening is a fusion technique used to generate imagery with a high spectral and spatial resolution (Figure 9b,c).

After preprocessing, feature detection and matching are the next steps. Strong features are so unique that they are, by description, outstanding and exceptional. Corners are good candidates and segmentation can help to classify and remove week features. However, this process decreases the number of features significantly [7]. Therefore, multiple feature detector operators are applied to increase the number of features [3]. Vegetation features often decrease the feature matching performance, especially when dealing with remotely sensed images, because of their dynamic appearances and characteristics in different images, even with only insubstantial imaging parameters. Therefore, it may be helpful to exclude vegetation features from the image matching process during the point cloud generation processes. In this regard, using the NDVI as an index to separate vegetated and non-vegetated areas is a viable and convenient approach and should improve the performance of image matching and the resultant point cloud data. The identification of features within dense vegetated areas based on the NDVI is demonstrated as an example in Figure 10. Firstly, the CIR images clearly show the vegetation within the area around the target stadium in red (Figure 10a). The calculated NDVI (b) with legend emphasizes the robustness and usefulness of performing vegetation detection with the NDVI. The final image (c) is a composite image of the RGB image, in which the vegetation is marked in green and the SURF features classified as vegetation are marked as yellow asterisks.

The final stage of the two-view SfM point cloud reconstruction is detecting and describing features with the SURF, FAST (BRIEF) and ORB algorithms. Points are removed if they are located in the vegetation areas. Afterwards, the remaining non-vegetated feature points are checked for outliers and these are removed as well (masking). This masking is accomplished over H-matrix estimation. Then, the next step is feature matching with Brute Force with respect to the type of descriptor. Finally, pose estimation and triangulation, including essential and fundamental matrix estimation with the remaining features, are performed to generate 3D point clouds.

## 4. Results and Discussions

### 4.1. Training and Model Testing

The training strategy and model selection process, as described in Section 3.3.2, is carried out in a stepwise procedure. The trained model is generated cross-sensor-wise and based on four categories. The validation is accomplished with a Confusion Matrix (CM) to evaluate the performance of the selected model with NDVI classification and is summarized in Table 7. The selected model is generated with 15,360 tiles at epoch 64. In addition, 10% or 1536 of each category are randomly selected for category-wise and balanced validation.

As listed in the table, the results of the two pansharpened sets (PAN74, PAN93) are more accurate than the multi-spectral ones by 35.9% (MS74, MS93). The two pansharpened sets achieve accuracies of 0.9466 and 0.9024 and kappa coefficients of 0.8932 and 0.9110, respectively.

### 4.2. Color Infrared Simulation Validation

A Pleiades satellite image was used to validate the effectiveness of the developed model for simulating color infrared images (Figure 11 and Figure 12). The target consists of a stadium and its environment (as shown in Figure 5). A larger sub-image (Figure 12a) covering different types of vegetation and vegetation arrangement was also tested. The quadratic tile mainly focuses on the stadium and close environment. The test images were excluded in the training phase when developing the model. The original infrared band of the Pleiades images can be treated as ground truth (GT) to evaluate the predicted (simulated) CIR result.

The predicted CIR, NIR and NDVI images and the ground truth of the target and its vicinity are first used for visual comparison and to validate the CIR simulated results (Figure 11). As shown in Figure 11, by comparing the predicted CIR (b) and the ground truth (c), the details are overall preserved. However, some surfaces like the solar panels or flat roof tops have a slight red discoloration instead of being solid gray. The NIR band (d) extracted from the predicted CIR exhibits slightly blurred edges compared to the ground truth (e). The comparison of the NDVI prediction (f) and ground truth (h) shows an overestimation of vegetation, especially areas with narrow areas of vegetation close to non-vegetated areas, like streets and trees close to the street.

The larger sub-image (Figure 12a) covers different landcover types; these are, from left to right, smaller buildings, a major highway, a park area with a stadium, supply facilities for the stadium with solar panels, smaller roads and high-rise buildings with a parking area, etc. The predicted CIR and the subsequent extracted NIR and NDVI calculated from the simulated CIR image all seem to have higher intensity values than the ground truth, but the prediction still preserves the overall trend and details. A histogram investigation is conducted to further quantitively analyze the simulation performance.

### 4.3. Histogram evaluation

Histograms provide, in contrast to the visual validation of Section 4.2, a quantitative comparison of the predicted (simulated) images and ground truth. The histogram itself provides information about the intensity range and distribution (CloudCompare, version 2.12.1). The comparison of the ground truth and prediction emphasizes their similarities and differences beyond a visible inspection and provides the opportunity for a more comprehensive analysis.

The predicted CIR (and extracted NIR) and the ground truth around the target stadium building and its vicinity are displayed in Figure 13. Each tile of the NIR and NDVI images simulated from the original Pleiades multi-spectral (MS) bands and its ground truth (Figure 13a,b) displays 256 × 256 pixels. By visual comparison, it appears that the NDVI calculated from the predicted NIR is slightly blurred compared to the one calculated from the original. In addition, the simulated image seems to have higher NDVI values (brighter) than the ground truth. However, analyzing the histograms shown in Figure 13c reveals that both histograms have similar distributions but in different ranges. Figure 13d–f display the NDVI ground truth images and the simulated counterparts predicted from a PAN Pleiades image divided into nine tiles. The prediction shows clear border noise, in addition to an increase in the intensity values of the vegetation. The histogram comparison indicates that although the ranges of the ground truth and predicted NDVI are different, their distribution patterns are similar.

The comparison of the ground truth and prediction with histograms and visual inspections of the CIR, NIR and NDVI from MS and PAN pansharpened the image subsets with similar scene contents (Figure 14). Firstly, MS in Figure 14I–III displayed three different scenes with a size of 256 × 256, showing the CIR, NIR and NDVI for the ground truth (GT) and prediction (P), in addition to the histogram. The first row shows a rural area mainly consisting of different types of vegetation and a water stream. The vegetation is captured well and is similar to the ground truth; however, the river seems to be blurred in the simulated image. The second row shows a similar number of buildings and vegetation (agriculture). The areas with a high chlorophyl content (bright red) are identified best, whereas low or non-chlorophyl areas are blurred and miss-classified as vegetation. Row III shows more buildings, and the prediction gets closer to the ground truth; however, it is still blurry. In the PAN cases, row IV, V and VI show three examples simulated from the PAN-sharpened (s) to (jj). Row IV shows a rural area covered partially by a cloud. Interestingly, the prediction enhances the contrast, so the prediction shows much more clear features than the ground truth. This contrast enhancement and the benefits are shown in row V as well. Row VI follows that trend too; however, it has difficulty predicting the natural curvy water stream entirely.

The examples presented above demonstrate that the proposed machine learning model can effectively simulate color infrared (CIR) images from RGB images; therefore, the NIR band can be extracted to calculate the NDVI for the segmentation of vegetation-covered areas. This can be very useful for images acquired using sensors or cameras without infrared channels such as regular UAV images. The comparison between the predicted NDVI and the ground truth calculated using the original infrared band of Pleiades very high-resolution satellite images (both multi-spectral and pansharpened versions) further demonstrates that the NDVI values calculated from the generated CIR images are highly correlated to the true NDVI values and should be capable of identifying vegetation-covered areas.

### 4.4. Structure from Motion on UAV Prediction and Application

The proposed two-stage workflow has an overall aim of improving 3D point cloud generation for static targets such as buildings from RGB images, especially UAV images. The developed two-stage point cloud reconstruction framework was tested on a real dataset collected using a regular RGB camera onboard a DJI UAV. The UAV data consist of a target building occluded by trees. As mentioned previously, detecting and masking vegetation-covered areas should help to improve 3D point cloud generation. Since there are no infrared channels in the test UAV dataset, the developed image-to-image translation procedure was applied to simulate the CIR images of the dataset. Then, the predicted NIR band was extracted to calculate NDVI values for segmenting vegetation areas from the scenes before the SfM operation for 3D reconstruction.

In general, the segmentation-driven SfM algorithm starts with feature detection and matching with respect to the NDVI-based vegetation-masked images. The vegetation segmentation and masking reduce weak features, whereas the application of multiple feature detector operators, namely SURF, ORB and FAST, increases the number of useful features. Then, feature matching with Homography matrix masking is performed to preserve the quality of the matched features. Finally, the last task is pose estimation and triangulation to generate point clouds.

The UAV scenes shown in Figure 15 display the prediction of CIR (b) from RGB (a), with padding on the bottom and right of the image. The extracted NIR (c), with removed padding, and the NDVI (d) are calculated based on the extracted NIR and original red band. The CIR shown in Figure 15b is a mosaic of 40 tiles (including the padding), which were originally divided for computing efficiency. All three translated images (b-d) seem to exhibit a clear “edge enhancement” effect and noise related to the padding, which are side effects of the CIR simulation. Figure 15e,f provide an isolated and closer look at the area marked by the orange box in (a) as two 256 × 256 titles. It is also noted that the illumination conditions, shadows, and colors can affect the CIR simulation result. For example, the green wall painting on the right building in (e) was mistakenly treated similarly to the trees (f).

Initially, an average of 149,181 features were detected (Table 8). After vegetation segmentation, an average of 133,808 features remained. This led to a difference of 10.31%, or 15,373.5 features. In other words, about 10% of feature points were removed, but the remaining features were more robust for image matching.

The results of the two-view SfM reconstruction in direct comparison are shown in Figure 16, without segmentation in Figure 17 and with segmentation applied in Figure 18. Without vegetation segmentation are shown in Figure 17a as a sparse point cloud and (b) as a point cloud colored by elevation. The density analysis and its histogram are, respectively, displayed in Figure 17c,d. The accumulated numbers of points are listed in Figure 17e. It was also noticed that among the reconstructed sparse point cloud, only 26,673 of the target points were partially occluded by vegetation. The initial point cloud needed the removal of 65.76% or 51,236 points to produce a “cleaner” point cloud. The Point Cloud colored by elevation emphasizes the influence and the level of occlusion that was, in this case, caused by vegetation. The density analyses shown in (c) and (d) were accomplished using the number of neighbors with a radius of 0.5 m for each point, respectively. The histogram shows the counts of 40 classes of the number of neighbors. The vegetated areas, some corners of the building and the satellite antenna show higher point density. However, the majority of the target, the building, show an overall low point density, as the histogram underlines.

The results of the two-view SfM reconstruction with vegetation segmentation are shown in Figure 18a as sparse point clouds, (b) as a point cloud colored by elevation, (c) the density analysis and (d) its histogram. The accumulated numbers of features detected without and with vegetation segmentation are summarized in Table 8. In addition, a comparison of without NDVI and with NDVI is summarized in Table 9, including the initial, the cleaned and the difference between the generated and processed point clouds. As expected, the number of points with the NDVI-based vegetation removal process is smaller than without. Likewise, the amount of noise manually removed is similar. However, the quantitative density analysis and visual analyses both show that the two-view SfM reconstruction can produce more plausible results on images treated with the vegetation segmentation and removal process, further demonstrating the advantage of the approaches proposed in this study.

Finally, the direct comparison shown in Figure 16 emphasizes the benefits of the framework proposed in this study. It can be seen that the occlusion of vegetation dominates in the foreground (a) of the scene, which took significant computational resources but also may have had a negative impact on the dense matching operation. Furthermore, it can be observed that the vegetation has a higher point density than the target (building). On the other hand, with the NDVI data generated in the proposed framework, the occlusion caused by vegetation is significantly reduced. The resultant point cloud in Figure 16b shows more clear building edges and corners, not to mention that computational resources were spent on the actual targets rather than on unwanted objects.

## 5. Conclusions and Suggestions

This study successfully enhanced a building point cloud from RGB UAV data with machine-learning-based image-to-image translation and two-view SfM algorithms. The developed machine learning procedure effectively translates original RGB UAV images to color infrared in order to help segment and remove vegetation from the original images to improve building point cloud generation. The implementation of the two-stage framework consisted of three fundamental steps, which included using resource-aware cross-sensor training to generate a robust model for RGB to color infrared translation, model verification and performance analysis, and image segmentation with the newly generated data to improve SfM.

The cross-sensor training and model selection used to train the machine learning model to predict CIR from RGB is a conditional GAN (Pix2Pix). The predicted CIR is then used to extract NIR and red bands, in order to calculate the NDVI. The proposed framework takes several key elements into consideration. Firstly, the relation between the number of categories and time shows that with less categories and less tiles, less time and resources are needed; meanwhile, the performance is kept stable. This leads to the application of fewer categories and a cross-sensor training idea. The cross-sensor training enhances the applicability, as well as increases the model’s robustness and performance. The evaluation of the results shows that the predicted CIR results can reach binary classification accuracies of 0.9466 and 0.9024 and have kappa coefficients of 0.8932 and 0.9110, respectively. Histogram-based evaluation demonstrates that the predicted NIR band is consistent with the original NIR data of the satellite test dataset.

The proposed algorithm and the developed machine learning model were applied to an UAV RGB image dataset without ground truth. The UAV dataset lacks the NIR band, meaning that the image-to-image translation of RGB2CIR fills that gap. The predicted CIR was then incorporated to generate 3D point clouds of a building target with a two-view SfM. The selected model successfully translated RGB to CIR data. The prediction and NDVI-based vegetation segmentation show reasonable results with minor padding and shadow noise issues. Nevertheless, the reconstruction shows decreased density in vegetated areas and the increased density of the target. The result also demonstrates that the proposed vegetation-driven two-view SfM framework can effectively reduce the influence of vegetation and generate more adequate point cloud data of the targets.

The cross-sensor training offers a variety of applications. However, data availability and copyright can be a challenge. In addition, the computational resource requirements and time can limit the applicability of the proposed approach. Nevertheless, the proposed two-stage two-view SfM reconstruction with vegetation segmentation and removal based on simulated NDVI values is still a viable and effective solution. The proposed solution improves the quality of the 3D building point cloud data generated from regular RGB images using SfM algorithms and does not require sophisticated sensors or expensive equipment. It is notable that the resolution gap of the cross-sensor training sample and the quality of the prediction are related. It is recommended that the cross-senor training is carried out carefully.

Additional points are worth investigating and incorporating into the proposed framework to further enhance the model’s performance in the future. For example, the further fine tuning of the categories of training images for machine learning is recommended, including training with a clear progression of high to low-resolution images. The use of a post-processing step to handle the boarder noise would also be interesting. With a focus on vegetation categories, the additional processing of green objects and shadows or illumination changes could further improve the model’s performance. In addition, the shadows cast by clouds are considered as occlusions; therefore, the removal of clouds and shadows would further improve point cloud generation with SfM algorithms.

In terms of motion-based scene structure reconstruction, the application of epipoles rather than only relying on features may provide a faster and more stable pose estimation [57]. On top of that, normalization often decreases the resolution of images, especially for satellite data. The application of deep features like SuperPoint in combination with SuperGlue may overcome this issue [58]. SuperPoint is a self-supervised machine learning operator that seems to be a convenient and adjustable solution. Alternatively, correspondence can be established with an AI scoring network [59] or pruning performed with global texture [60].

## Figures and Tables

**Figure 1 sensors-24-02358-f001:**
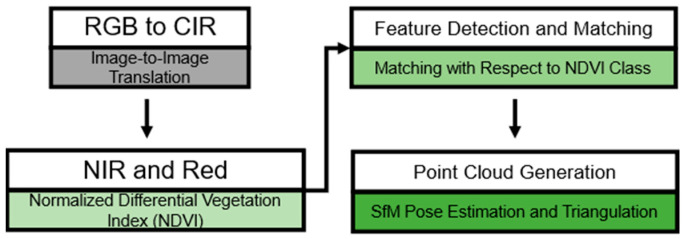
Two-stage framework separated into the machine learning technique (gray) and the application in three steps (green). Firstly (gray), the CIR image is generated from RGB with image-to-image translation. Then (light green), the NDVI is calculated with the generated NIR and red band. Afterwards, (medium green), the NDVI segmentation and classification is used to match the detected features accordingly. Finally (dark green), pose estimation and triangulation are used to generate a sparse 3D point cloud.

**Figure 2 sensors-24-02358-f002:**
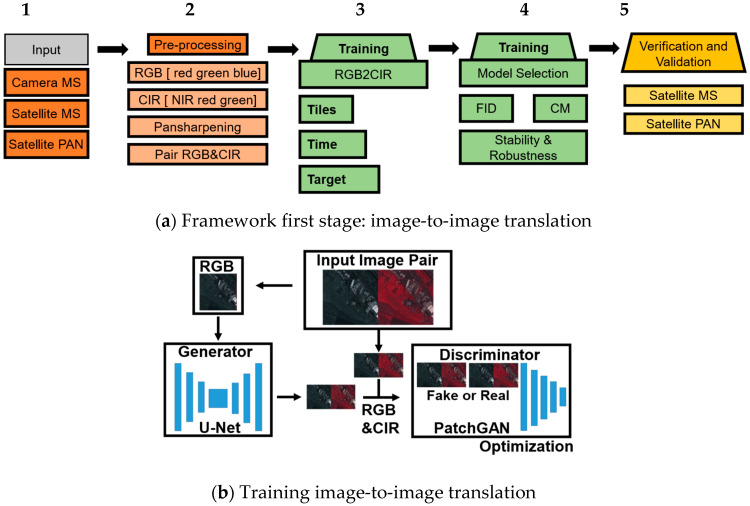
First stage of the two-stage workflow. (**a**) Image-to-image translation in 5 steps for RGB2CIR simulation. In general, input and pre-processing (orange), training and testing (green) and verification and validation (yellow) (**b**) Image-to-image translation training.

**Figure 3 sensors-24-02358-f003:**
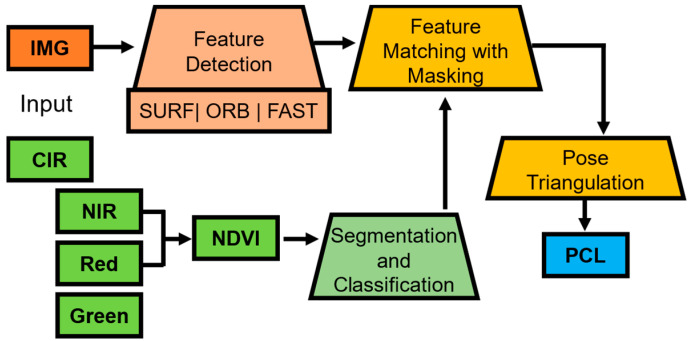
Framework second stage: segmentation-driven two-view SfM algorithm. The processing steps are grouped by color, the NDVI related processing (green), the input, feature detection (orange), feature processing (yellow) and the output (blue).

**Figure 4 sensors-24-02358-f004:**
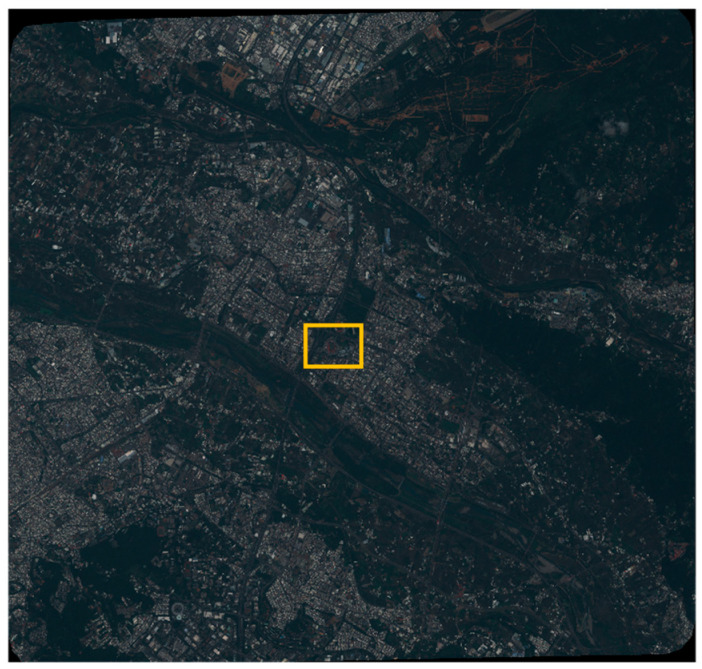
Pleiades VHR satellite imagery, with the nadir view in true color (RGB). The location of the study target is marked in orange and used for validation (see Section 3.2.3).

**Figure 5 sensors-24-02358-f005:**
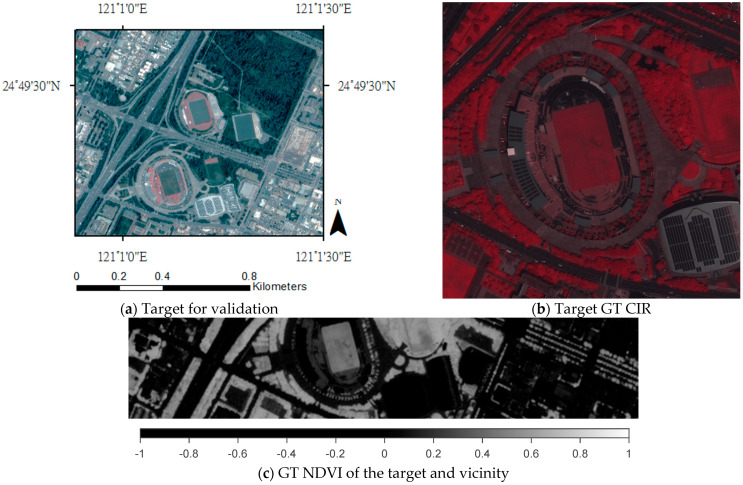
The target for validation captured by Pleiades VHR satellite. (**a**) The target stadium; (**b**) the geolocation of the target (marked in orange in Figure 4); (**c**) the target ground truth (GT) CIR image. GT NDVI of the target building and its vicinity.

**Figure 6 sensors-24-02358-f006:**
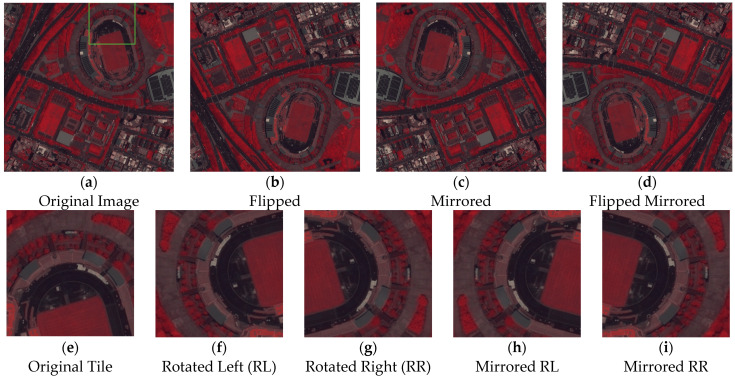
Morphological changes on the image covering the target and image tiles. (**a**) Original cropped CIR image of Pleiades Satellite Imagery (1024 × 1024 × 3). A single tile, the white rectangle in (**a**), is shown as (**e**). (**b**–**d**) and (**f**–**i**) are the morphed images of (**a**) and (**e**), respectively.

**Figure 7 sensors-24-02358-f007:**
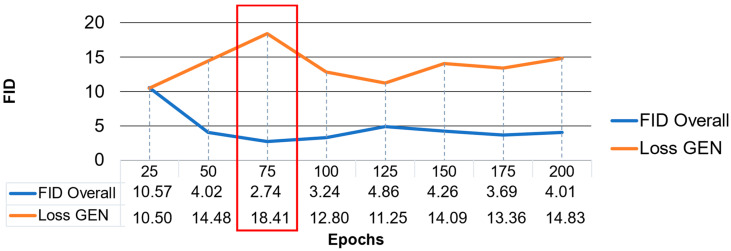
Training over 200 epochs for model selection. The generator loss (loss GEN) plotted in orange and, in contrast, FID calculation results in blue.

**Figure 8 sensors-24-02358-f008:**
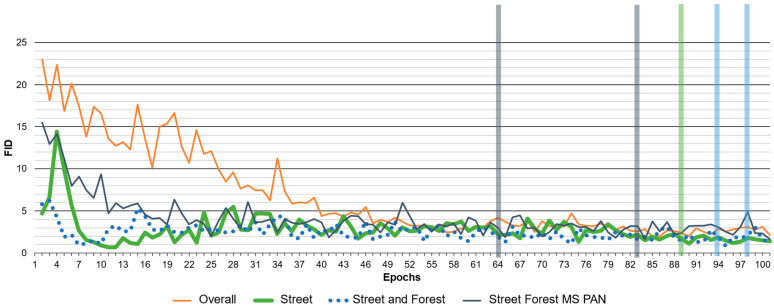
Training Pix2Pix for model selection with FID. The epochs with the best FID and CM are marked for every test run, expect overall, with colored bars respectivly. The numbers are summarized in Table 5.

**Figure 9 sensors-24-02358-f009:**
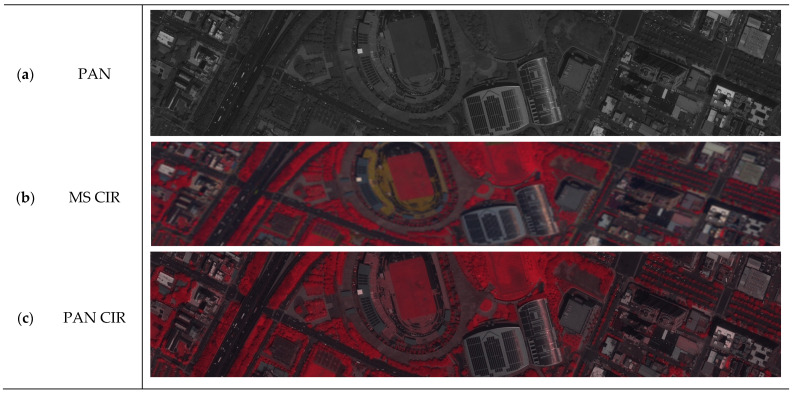
CIR pansharpening on the target. The high-resolution panchromatic image is used to increase the resolution of the composite CIR image while preserving spectral information. From top to bottom, (**a**) panchromatic, (**b**) color infrared created from multi-spectral bands, and (**c**) pansharpened color infrared are shown.

**Figure 10 sensors-24-02358-f010:**
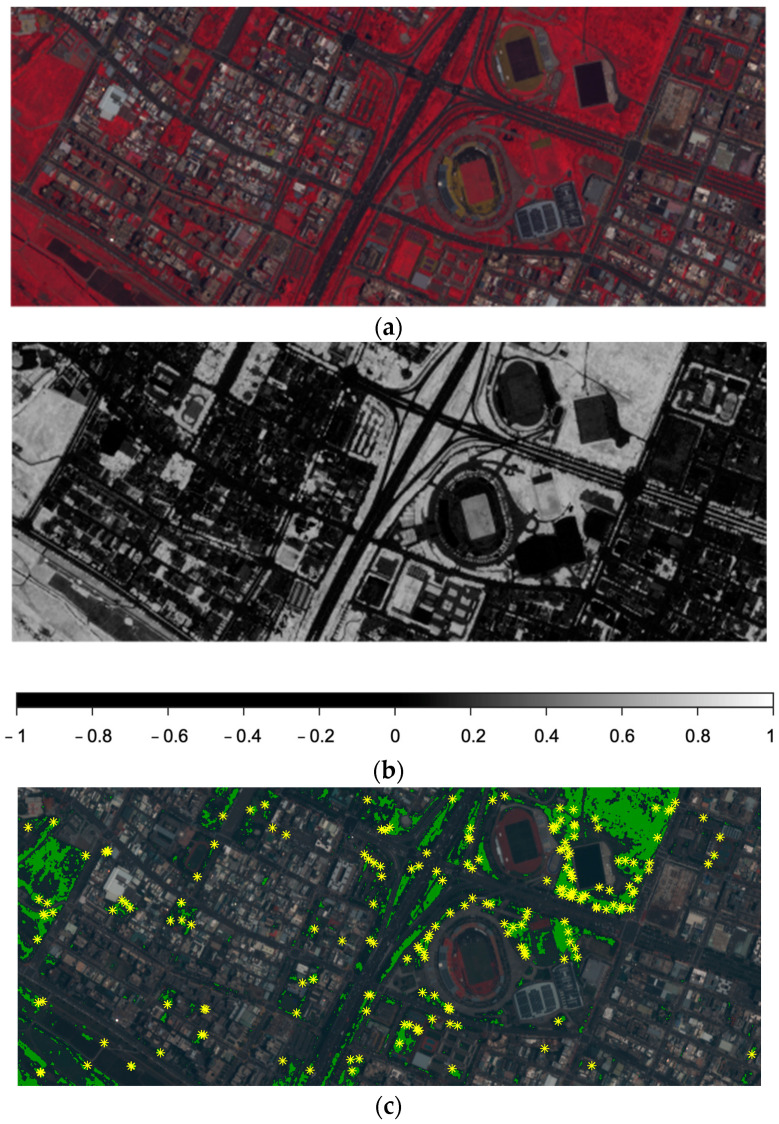
Example of vegetation feature removal to the north of the stadium. (**a**) CIR images; (**b**) NDVI image with legend; (**c**) identified SURF features (yellos asterisks) within dense vegetated areas (green) using 0.6 as the threshold.

**Figure 11 sensors-24-02358-f011:**
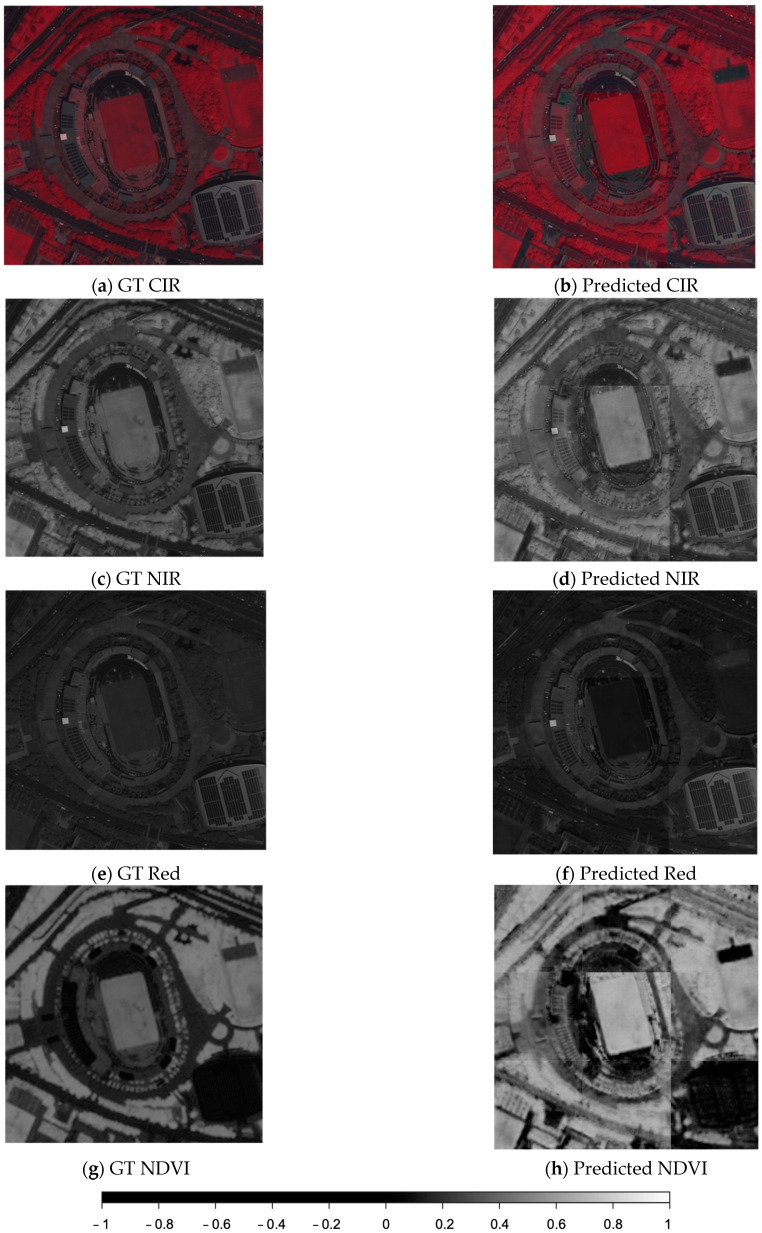
Comparison between the prediction and the ground truth (GT) of the CIR, NIR and NDVI (incl. legend) of the main target (a stadium) and vicinity.

**Figure 12 sensors-24-02358-f012:**
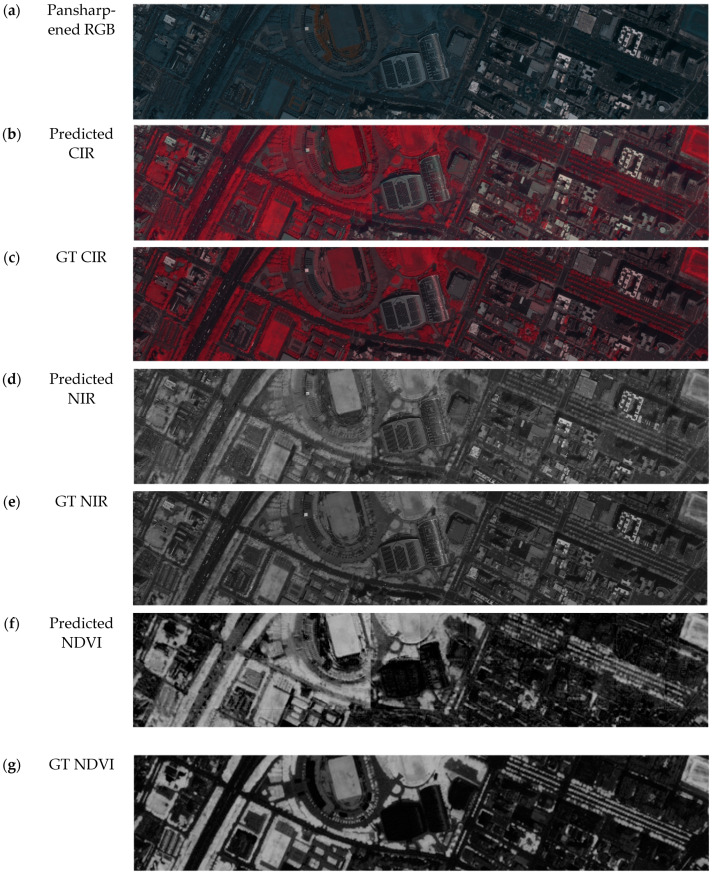
Comparison between the prediction and the ground truth (GT) of the CIR, NIR and NDVI generated from a pansharpened RGB satellite sub-image.

**Figure 13 sensors-24-02358-f013:**
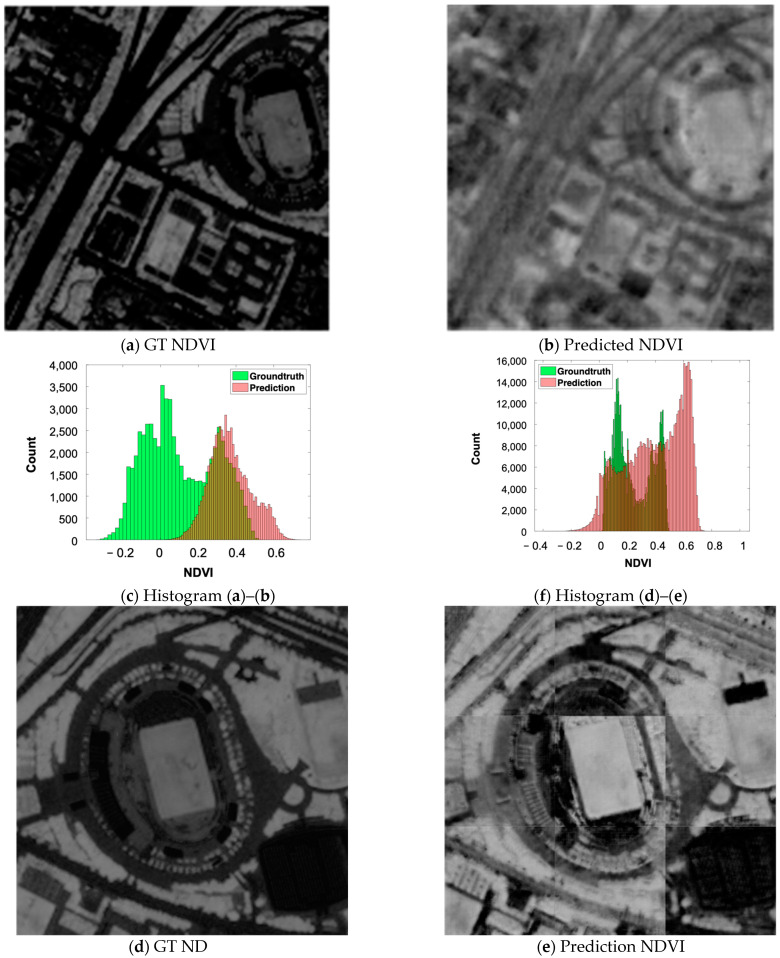
Histogram and visual inspection of the CIR and NDVI simulated using MS and PAN images on the target stadium. (**a**–**c**) Ground truth (GT) and NDVI predicted using one tile with the size of 256 × 256 from MS Pleiades and their histograms. (**d**–**f**) Ground truth of CIR, NIR, NDVI and predicted NIR and NDVI images from nine tiles of the PAN Pleiades images and histograms for NDVI comparison.

**Figure 14 sensors-24-02358-f014:**
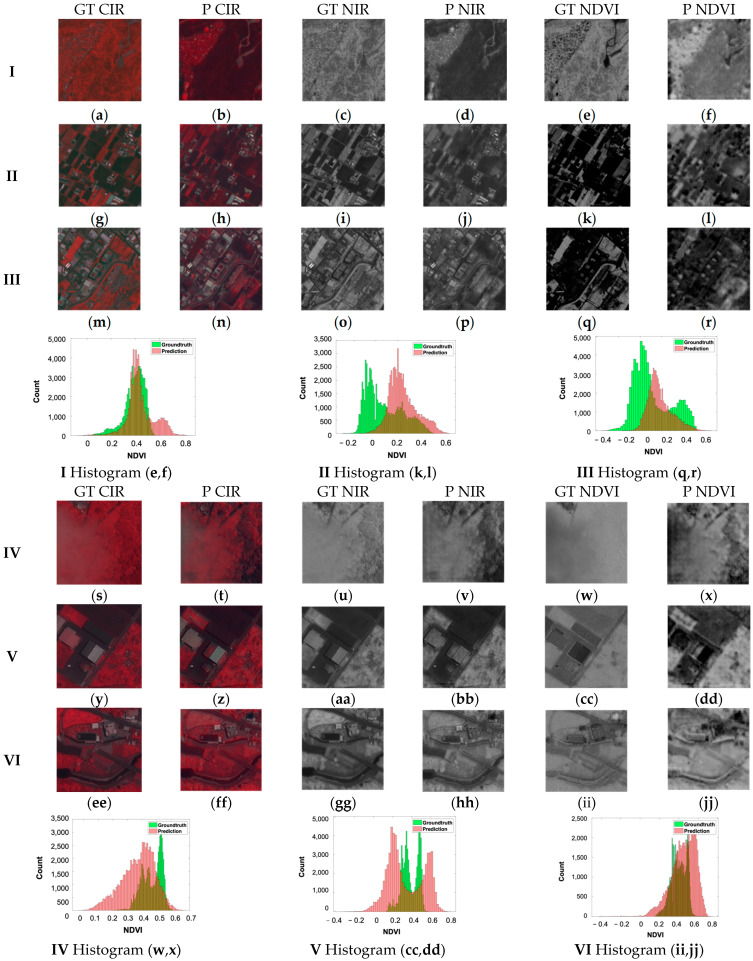
Histogram and visual inspection of MS (**I**–**III**) and PAN (**IV**–**VI**) examples of Zhubei city.

**Figure 15 sensors-24-02358-f015:**
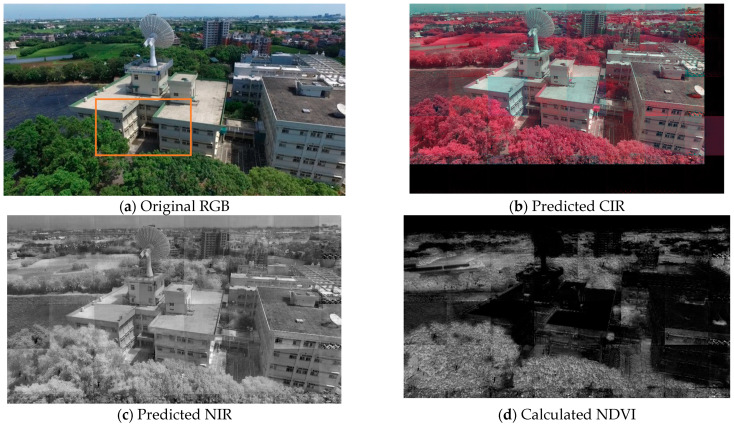
Prediction of CIR, NIR and calculated NDVI of a UAV scene: (**a**) RGB, (**b**) predicted CIR image, (**c**) the extracted NIR band of (**b**), and (**d**) calculated NDVI with NIR and red band. A close-up view of the area marked with an orange box in (**a**) is displayed as two 256 × 256 tiles in RGB (**e**) and the predicted CIR (**f**).

**Figure 16 sensors-24-02358-f016:**
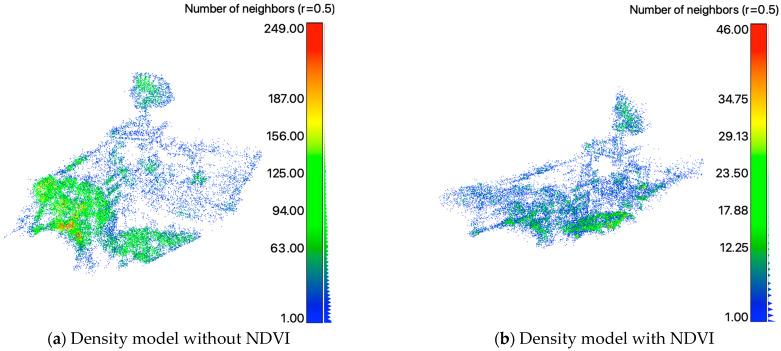
Direct comparison between without (**a**) and with vegetation segmentation (**b**). Areas of low density shown in blue, areas of high density shown in red.

**Figure 17 sensors-24-02358-f017:**
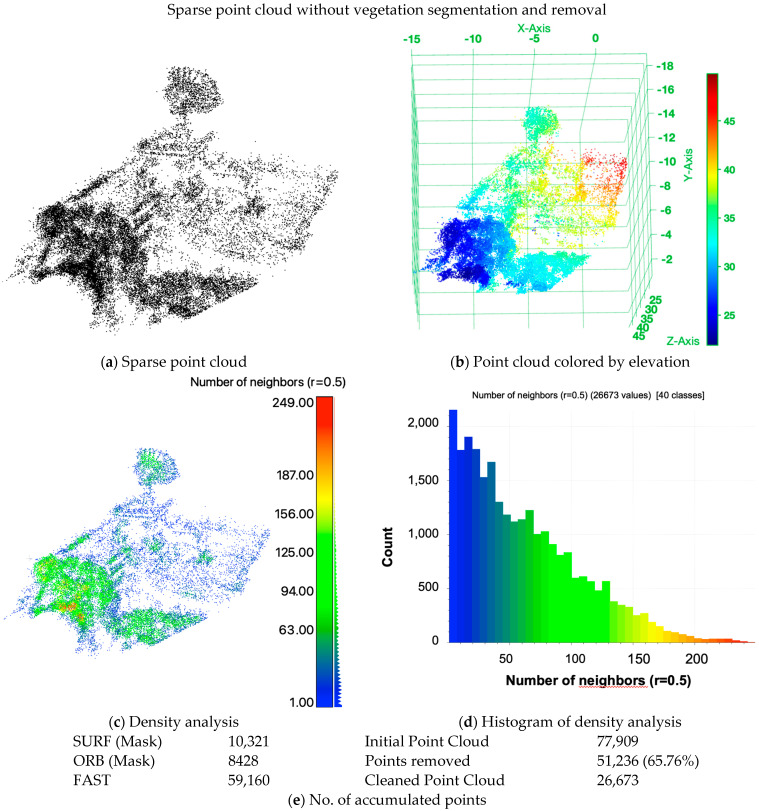
Two−view SfM 3D sparse point cloud without the application of NDVI−based vegetation removal on the target CSRSR. (**a**) Sparse point cloud with no further coloring; (**b**) point cloud colored by elevation; (**c**) density analysis and the corresponding histogram (**d**). In addition, Table (**e**) shows the accumulated number of points over the three operators (SURF, ORB and FAST) and the initial and manually cleaned and processed point cloud.

**Figure 18 sensors-24-02358-f018:**
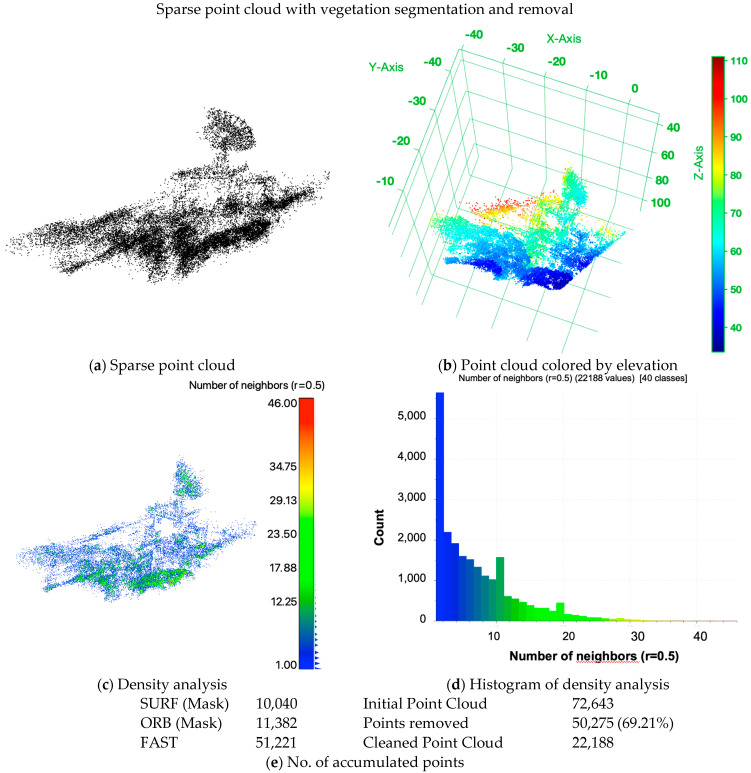
Two−view SfM reconstructed 3D sparse point cloud with vegetation segmentation and removal process based on simulated NDVI of the target building. (**a**) Sparse point cloud with no further coloring; (**b**) point cloud colored by elevation; (**c**) density analysis and (**d**) the histogram. In addition, (**e**) lists the accumulated number of points over the three operators (SURF, ORB and FAST) after segmentation, with 0.5 NDVI as the threshold to mask vegetation in SURF and ORB, and the initial and manually cleaned point cloud.

**Table 1 sensors-24-02358-t001:** EPFL RGB-NIR scene dataset. The categories in alphabetic order. In addition, the total number of images (RGB + NIR) and the number, respectively, of RGB and NIR.

Category		Category		Category	
Country	104 (52)	Indoor	112 (56)	Street	100 (50)
Field	102 (51)	Mountain	110 (55)	Urban	116 (58)
Forest	106 (53)	OldBuilding	102 (51)	Water	102 (51)

**Table 2 sensors-24-02358-t002:** Pleiades tri-stereo pairs including the view, the acquisition data and incident angle.

View	Date	Incident Angle
Nadir	2019-08-27T	12.57°
	9:18:44.807	
Forward	2019-08-27T	17.71°
	08:43:17.817	
Backward	2019-08-27T	16.62°
	08:47:06.900	

**Table 3 sensors-24-02358-t003:** The pre-processed available images. The Pleiades dataset consists of two groups, the multi-spectral (MS) and the panchromatic (PAN). The MS and EPFL data are processed with morphological changes (Figure 6) to generate more tiles for training.

Pleiades	MS					PAN		
	Size	Sliced	Tiles 1	Tiles 2	Total	Size		Sliced
Img1 (70)	5285 × 5563	462	1386	7392	9240	21,140 × 22,250		7221
Img2 (74)	5228 × 5364	441	1322	7498	8820	20,912 × 21,452		6888
Img3 (93)	5189 × 5499	462	1386	7392	9240	20,756 × 21,992		7052
EPFL								
Images	Tiles 1	Tiles 2		Tiles total				
369	4452	17,808		22,260				

**Table 4 sensors-24-02358-t004:** EPFL RGB-NIR dataset and training variation. There are 22,260 training tiles for 9 categories and around 50 h of training for 100 epochs. The table also shows training with only one (Streets) category (Training 1) and two (Streets and Forest) categories (Training 2). Tiles 1 and Tiles 2 are the products of the morphological changes described in Figure 6.

EPFL	Training 9	Training 1	Training 2
Images	369	16	32
Tiles 1	4452	240	480
Tiles 2	17,808	768	1536
Tiles Total	22,260	3840	7680
Time	50 h 48 min	6 h 42 min	13 h 1 min

**Table 5 sensors-24-02358-t005:** Training and testing for model selection.

	Overall	Streets	Streets, Forest	Streets, Forest, MS and PAN
Training Tiles	22,260	3840	7680	15,360
Numb. Categories	9	1	2	4
Time	50 h 48 min	6 h 42 min	13 h 1 min	25 h 51 min
EPFL RGB-NIR	9 categories	1 category	2 categories	2 categories
MS	×	×	×	1 image
PAN	×	×	×	1 image

**Table 6 sensors-24-02358-t006:** FID summary and CM for training model selection.

	Street	Street and Forest		Street, Forest, MS and PAN	
Model	89	98	94	83	64
CM					
Accuracy	0.8136	0.5781	0.7645	0.6830	0.7125
Precision	0.7285	0.5423	0.6798	0.6120	0.6349
F1-score	0.8429	0.7033	0.8093	0.3660	0.4251
Kappa	0.6273	0.1563	0.5290	0.7593	0.7767
FID	1.132	3.057	0.839	1.663	1.905

**Table 7 sensors-24-02358-t007:** Model testing for validation separating MS and PAN results. Pleiades MS and PAN, respectively, for the Forward (74) and Backward (93) view, including the average (AVG). In addition, the absolute Difference (Diff abs) and relative difference (Diff %) used to compare MS and PAN.

Pleiades	Accuracy	Recall	F1-Score	Kappa
MS74	0.5886	0.5486	0.7085	0.1772
MS93	0.5956	0.5528	0.7120	0.1912
AVG MS	0.59	0.55	0.71	0.18
PAN74	0.9466	0.9035	0.9493	0.8932
PAN93	0.9024	0.8366	0.8048	0.9110
AVG PAN	0.92	0.87	0.88	0.90
Diff abs	0.33	0.32	0.17	0.72
Diff %	35.9	36.8	19.3	80

**Table 8 sensors-24-02358-t008:** Number of points without or with the processing vegetation segmentation using the NDVI calculated using the NIR and red bands of the CIR image simulated from the original RGB UAV scene.

Features	Initial	After	Difference	
Image 1	148,717	133,186	15,531	10.44%
Image 2	149,646	134,430	15,216	10.17%
AVG	149,181	133,808	15,373.5	10.31%

**Table 9 sensors-24-02358-t009:** Initial and cleaned sparse point cloud with and without NDVI-based vegetation segmentation and removal process.

PCL	without NDVI		with NDVI		Difference	
Initial	77,909		72,643		5266	
Difference	51,236	65.76%	50,275	69.21%	781	15.83%
Cleaned	26,673		22,188		4485	

## Data Availability

The RGB-NIR Scene Dataset from École Polytechnique Fédérale de Lausanne, CH, online available: https://ivrlwww.epfl.ch/supplementary_material/cvpr11/index.html (accessed on 6 April 2024). UAV dataset not applicable. Pleiades dataset not applicable.

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
