# Peer review of "Enhancing Building Point Cloud Reconstruction from RGB UAV Data with Machine-Learning-Based Image Translation"

_sensors, 2024, doi:10.3390/s24072358_

Round 1

Reviewer 1 Report

Comments and Suggestions for Authors

This paper proposes a two-stage framework using machine learning image translation to generate near-infrared (NIR) images from RGB images captured by UAVs. The goal is to improve the accuracy of 3D point cloud reconstruction by reducing noise caused by vegetation, which can be effectively identified using NIR data. This is an interesting research. Here are some comments for you.

1. In Abstract, please summarize your research in a nutshell, emphasizing the big takeaways and why they matter. Currently, you put too many efforts on background.

2. The introduction sets the context for the study and identifies the significance of the research. However, it could be improved by providing more background information on the current state of knowledge and research gaps in the field. You may introduce more examples for machine learning in civil engineering, such as 10.1016/j.autcon.2023.104849, 10.1016/j.measurement.2024.114190, 10.1016/j.autcon.2021.103678, and 10.1016/j.ymssp.2023.110532

Is there a similar study for the multiple contribution points of the manuscript? This question needs to be addressed in the introduction.

3. Regarding the data, how representative is the publicly available RGB-NIR dataset used for training?

Are there limitations in using satellite imagery for NDVI calculation compared to actual UAV-acquired NIR data?

4. How sensitive is the framework to the choice of feature detector operators? Are there alternative methods for vegetation removal that could be explored?

5. How is the accuracy of the generated 3D point cloud measured beyond visual inspection?

6. Could quantitative metrics be used to compare the quality of point clouds generated with and without the proposed framework?

7. Are there specific types of vegetation or environmental conditions where the framework performs poorly? Can the claimed accuracy of 0.9466 and 0.9024 be verified on independent datasets? How scalable is the framework for processing large datasets or complex environments?

Reviewer 2 Report

Comments and Suggestions for Authors

This study the author presents a two-stage framework to improve 3D point cloud reconstruction from images, particularly in areas with vegetation. The framework addresses the challenge of vegetation detection in images that lack near-infrared (NIR) data. The first stage involves cross-sensor training, model selection, and evaluation of image-to-image translation with Generative Adversarial Networks (GAN) for RGB to color infrared (CIR) conversion. The second stage involves feature detection, removal based on NDVI classification, masking, matching, pose estimation, and triangulation to generate a sparse 3D point cloud. The framework uses a publicly available RGB-NIR dataset, satellite, and UAV imagery. The study demonstrates the potential of machine learning in enhancing point cloud reconstruction from RGB UAV data by simulating color infrared images from original RGB bands for vegetation segmentation. Overall, the manuscript is well-composed. However, it would be beneficial to highlight the author’s contributions in the introductory section.

Reviewer 3 Report

Comments and Suggestions for Authors

The manuscript is good and structured, however find my little comments/suggestions to further improve its quality:

- The introductory section should end with the motivation and objective of the work. 

- Starting a heading/subheading is inappropriate. example (Line 61) 2. GAN.......

- Too many terminologies abbreviated that are not easy to comprehend

- Fig 1 is not mentioned in the manuscript

- What is the difference between section 3.1 and section 3.3?  Normally what we should have are materials and methods

- This is needed to distinguish your materials and methods from the Results section. There are a lot of statements that are disjointed in the section. Consider placing them appropriately

- Check some grammatical and spelling errors

Comments on the Quality of English Language

Minor revision required

Round 2

Reviewer 1 Report

Comments and Suggestions for Authors

 Accept in present form
